# Perceptions on artificial intelligence-based decision-making for coexisting multiple long-term health conditions: protocol for a qualitative study with patients and healthcare professionals

Niluka Jeewanthi Gunathilaka ,[1] Tiffany E Gooden ,[2] Jennifer Cooper,[2] Sarah Flanagan,[2] Tom Marshall,[2] Shamil Haroon ,[2] Alexander D'Elia,[2] Francesca Crowe ,[2] Thomas Jackson,[2] Krishnarajah Nirantharakumar ,[2] Sheila Greenfield [2]

NJG and TEG are joint first authors.

¹Ministry of Health Sri Lanka, Colombo, Sri Lanka
²Institute of Applied Health Research, University of Birmingham, Birmingham, West Midlands, UK

**Correspondence to**
Dr Francesca Crowe;
F.Crowe@bham.ac.uk

## ABSTRACT

**Introduction** Coexisting multiple health conditions is common among older people, a population that is increasing globally. The potential for polypharmacy, adverse events, drug interactions and development of additional health conditions complicates prescribing decisions for these patients. Artificial intelligence (AI)-generated decision-making tools may help guide clinical decisions in the context of multiple health conditions, by determining which of the multiple medication options is best. This study aims to explore the perceptions of healthcare professionals (HCPs) and patients on the use of AI in the management of multiple health conditions.

**Methods and analysis** A qualitative study will be conducted using semistructured interviews. Adults (≥18 years) with multiple health conditions living in the West Midlands of England and HCPs with experience in caring for patients with multiple health conditions will be eligible and purposively sampled. Patients will be identified from Clinical Practice Research Datalink (CPRD) Aurum; CPRD will contact general practitioners who will in turn, send a letter to patients inviting them to take part. Eligible HCPs will be recruited through British HCP bodies and known contacts. Up to 30 patients and 30 HCPs will be recruited, until data saturation is achieved. Interviews will be in-person or virtual, audio recorded and transcribed verbatim. The topic guide is designed to explore participants' attitudes towards AI-informed clinical decision-making to augment clinician-directed decision-making, the perceived advantages and disadvantages of both methods and attitudes towards risk management. Case vignettes comprising a common decision pathway for patients with multiple health conditions will be presented during each interview to invite participants' opinions on how their experiences compare. Data will be analysed thematically using the Framework Method.

**Ethics and dissemination** This study has been approved by the National Health Service Research Ethics Committee (Reference: 22/SC/0210). Written informed consent or verbal consent will be obtained prior to each interview. The findings from this study will be disseminated through peer-reviewed publications, conferences and lay summaries.

## STRENGTHS AND LIMITATIONS OF THIS STUDY

⇒ This study will use a UK population-based database comprising more than 30 million primary healthcare electronic records to facilitate the selection of a diverse sample of patients with multiple long-term conditions within the West Midlands in terms of key demographics and mix of co-occurring conditions.

⇒ Including perspectives from both patients and healthcare professionals will allow us to capture doubts, barriers or issues in using artificial intelligence (AI)-based clinical support tools from the people who will be affected the most by the development and implementation of an effective AI tool.

⇒ Use of a systematic methodology and extensive involvement of our Patient and Public Advisory Group have ensured our data collection tools are fit for purpose.

⇒ Despite the strength of using a UK population-based database, the database may also provide limitations due to possibilities of under-representation of certain groups due to structural inequalities.

⇒ Non-probability sampling and including patients only from the West Midlands may limit generalisability.

## INTRODUCTION

One quarter of adults in England have two or more long-term health conditions.[1 2] Having four or more long-term health conditions is strongly correlated with increasing age and has a significant contribution towards health service utilisation.[3] People with multiple long-term conditions are often prescribed multiple different medications (polypharmacy), which can lead to multiple side effects in addition to symptoms from their health conditions.[4]

Evidence suggests that polypharmacy can result in lower quality of life[5] and a threefold to fourfold increased risk in mortality among people aged 65 years or above.[6]

From the healthcare perspective, the more conditions a person has and more medications they take, the harder it is for healthcare professionals (HCPs) to consider all the factors when determining the best treatment plan, as clinical guidelines for one condition do not usually consider other existing health conditions.[7] Evidence is lacking on how to best treat health conditions for people with multiple long-term health conditions which in many cases, leads to excess treatment burden.[8] Due to limited resources for people with multiple long-term health conditions and the complex interaction between different disease conditions, primary care HCPs have identified the need for developing and adopting guidelines on how to best deliver and manage care for those with multiple health conditions.[9] Evidence-based guidelines are typically established from findings of clinical trials for individual diseases; however, such trials often exclude people with multiple health conditions.[7] As a result, HCPs are provided with guidelines for individual conditions that are often too complex and heterogenous to combine or integrate to determine the best treatment option for individuals with multiple health conditions. Observational studies could be conducted using large-scale population-based data (such as routinely collected electronic health records) to better understand the impact of medication on multiple health conditions; however, such studies are limited by the inability to account for individual patient medical histories, demographics (eg, age, sex, ethnicity) and the variability in decisions made by HCPs providing care.

Artificial intelligence (AI) tools are one solution that is being suggested to overcome some of the limitations in guidelines and evidence base for patients with multimorbidity.[10] AI is broadly defined as the science of "machines [that] do things that would require intelligence if done by people".[11] Machine learning is a type of AI in which a computer with self-learning capacity can generate predictive algorithms and identify patterns from data.[12] This has been successful at tackling complex problems outside of healthcare;[10] however, the capacity to process large amounts of information, systematically and reliably, faster than the human brain means there is considerable potential for AI in complex healthcare decision-making. The Royal College of General Practitioners has recognised that there is vast potential for AI in general practice.[13] There are some working examples to date in other settings (eg, secondary care, intensive care), where AI-based tools have been developed and validated for risk stratification and patient outcome optimisation,[14] but application of AI-tools in primary care is still in its infancy.[15 16]

The OPTIMising therapies, disease trajectories and AI-assisted clinical management for patients Living with multiple long-term health conditions (OPTIMAL) study aims to produce an AI-based tool to be used in primary care settings for planning the best treatment strategies and predicting the next health condition that people with multiple health conditions might develop to inform screening, investigations, prevention and/or treatment. Using population-based anonymised data available from primary and secondary healthcare records, the OPTIMAL study will determine the trajectories of disease accumulations for people with multiple health conditions and the contribution of medicines on the trajectories. The team will then develop a predictive model for the next likely disease and the best treatment option in the context of multiple conditions and multiple treatment options and incorporate this into an AI-based decision-making tool that can be used by HCPs and patients for joint decision-making for the best treatment plan.

For AI decision-making tools to be successfully implemented and effectively used in clinical practice, it is important that HCPs and patients trust, understand and see the value of using the tool. A 2021 systematic review of 23 qualitative studies exploring patient and public perspectives towards AI in the clinical setting found that the public were broadly positive about the concept of AI, but were concerned about the effectiveness of AI tools and felt that implementation should have human oversight.[17] However, in three-quarters of the included studies, discussions were based on AI as a hypothetical concept, rather than a real-world example (likely because few existed), which may limit the depth or specificity of the discussions.[17] Some qualitative studies have reported concerns from patients regarding the lack of a 'human touch' in using AI technology, but most have optimism that AI-based tools could free up HCP time, leaving more time for patient interactions.[18] Studies of clinicians' attitudes to AI are also generally positive, especially around the potential to take over mundane administrative tasks and synthesise data.[18–21] However, studies exploring UK general practitioners' (GPs) perspectives, including a large survey of 720 respondents, found that GPs believe human empathy, communication and tailoring to individual patients' values could not be replicated by AI.[21–23] Another study examining GP perspectives on using an AI-based documentation tool found they had concerns about the impact on their professional autonomy and who would bear the medicolegal responsibility for any outcomes.[19] Those in managerial and regulatory roles (including clinicians) have also expressed concerns over regulatory oversight of AI, the transferability/generalisability of the tools and the challenge of adopting AI within existing systems.[9] This was felt to be especially difficult within primary care which involves multiple independent practices and deals with a high burden of what were thought of as 'non-digitisable' healthcare problems such as mental health and chronic illness.[9]

Most existing studies exploring perspectives on AI have treated it as a broad and hypothetical concept.[17 18] Furthermore, there is limited evidence that describes patient perceptions of using AI-based tools for managing existing conditions or multiple health conditions.[21] Exploration of current and informed perspectives using a real-world

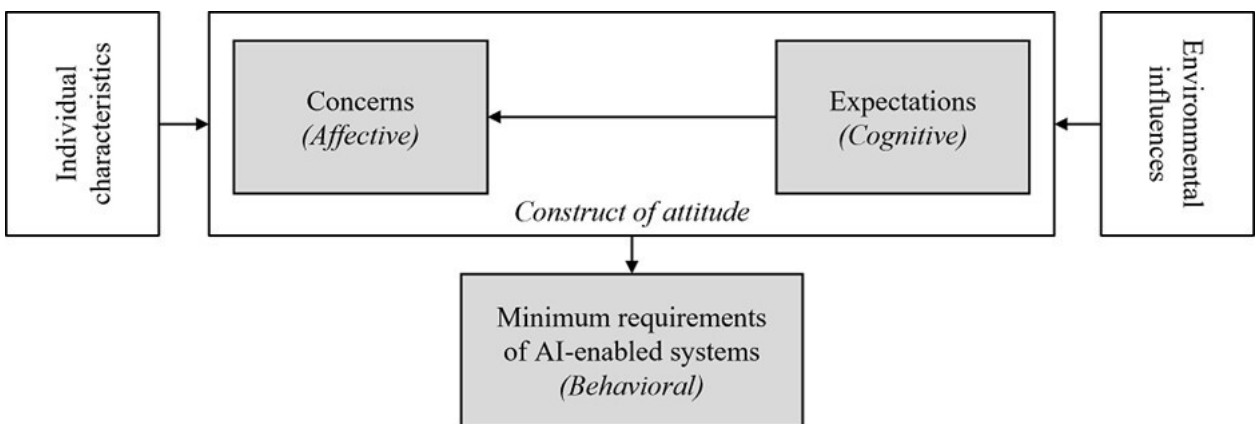

**Figure 1** Buck *et al*'s model of the general practitioners' determinants of attitudes towards AI-enabled systems This figure is reproduced under the terms of the Creative Commons Attribution License (https://creativecommons.org/licenses/by/4.0/) from work published in the Journal of Medical Internet Research,[25] available at https://www.jmir.org/2022/1/e28916. AI, artificial intelligence.

example of an AI-based tool is needed. Insights from HCPs regarding the challenge of implementing this technology in the management of multiple coexisting health conditions in primary care will be especially valuable.

Input from patients with multiple health conditions and HCPs on using the AI-based tool is critical in its development and implementation to ensure any specific barriers for the use of the tool can be appropriately addressed. Therefore, another component of the OPTIMAL study is a qualitative study which this protocol describes.

### Theoretical framework

Theoretical frameworks can be used at all stages in qualitative research to allow for a deeper understanding of how people and the cultures and organisations they are part of operate, interact and behave.[24] Several theoretical frameworks have been used to conceptualise attitudes and barriers to the introduction of AI in healthcare settings.[9 25 26] To explore the perspectives of GPs on AI in primary care in Germany, Buck *et al*[25] built on Rosenberg's[27] previous work on the construct of attitudes to change (figure 1). This model distinguishes the affective (emotional reactions, empathy and feelings), cognitive (ideas and knowledge) and behavioural (extent to which attitudes predict actions and intentions to act) dimensions to attitudes to use of AI-enabled systems in healthcare.[27 28] Individual characteristics such as age and prior experience with AI, and environmental influences such as the positive attitude from primary care professional bodies also influenced attitudes to AI. Given that the application of AI in primary care is largely hypothetical at present, current attitudes to AI from potential users are critical to the development of technologies that could then be implemented in practice. Therefore, we did not feel it appropriate to overly constrain the work by introducing theory, and therefore a particular slant, from the very outset. However, we will apply Buck *et al*'s theoretical lens at the analysis stage. This lens will be used as a framework to further explore the data from HCP interviews to provide a broader conceptual understanding of how the

affective, cognitive and behavioural components of attitude determine how HCPs evaluate their views on AI in managing multiple health conditions.

### Study aim

Through semistructured interviews (SSIs), this qualitative study aims to understand patients' and HCPs' perceptions of the advantages and disadvantages presented by AI-informed decision-making compared with clinician-directed decision-making for the management of multiple health conditions.

## METHODS AND ANALYSIS
### Study design and participants

This qualitative study will be conducted using SSIs[29] with two key groups of stakeholders. The first group will include people aged 18 years or older, with more than one long-term health condition, who are registered with a GP in the West Midlands County in England that contributes to the Clinical Practice Research Datalink (CPRD) Aurum database. Sixty-three conditions were considered when assessing eligibility (online supplemental material); these were derived through existing evidence[30] and discussions with the OPTIMAL Patient and Public Advisory Group (PPAG). Patients with a terminal diagnosis (prognosis of less than 12 months) will be excluded to avoid undue distress as well as those deemed by their GP not to have the capacity to consent or participate. For the second group, HCPs currently working in the UK involved in managing patients with multiple health conditions will be eligible to take part. A diverse range of professionals including but not limited to geriatricians, GPs, community pharmacists and specialist nurses will be eligible and invited to participate.

### Recruitment

CPRD Aurum is a population-based database of electronic primary healthcare records of anonymised data on diagnoses, tests, prescriptions and demographics,[31]

and will be used in the first instance to select eligible patients (figure 2). Prevalence and incidence data of many conditions in CPRD Aurum have been validated and demographics have been deemed representative of the general population in England.[31] Anonymised patient identifiers of patients from practices within the West Midlands will be accessed using the Data Extraction for Epidemiological Research (DExtER) software.[32] The eligible list of patients' unique CPRD identification numbers will be uploaded to CPRD's interventional research service platform (IRSP). CPRD will contact GP practices asking them to take part in the study. Initially 300 eligible patients from 10 GP practices will be selected and invited to participate. Each practice that takes part will be provided a list of eligible patients together with their CPRD ID number through the IRSP dashboard within their practice. Once this list has been received by the practices, either the practice manager, nurse or GP will view the eligible patients' details. They will review their medical history to ensure they meet the eligibility criteria. Once eligibility is confirmed, these patients will be sent an invitation letter and participant information sheet (online supplemental material) from their GP. Interested patients will be asked to call or email the OPTIMAL research team, at which point they can ask any questions about the study.

To augment this recruitment method for patients and ensure we get a diverse sample, we will also circulate recruitment flyers through voluntary, third sector and patient support and community groups, including but not limited to Citizens UK[33] and Birmingham Voluntary Services Council,[34] and through help from the OPTIMAL PPAG. Flyers will include information on eligibility and how to contact the OPTIMAL team to learn more about the study, ask questions and receive a copy of the participant information sheet before deciding if they would like to take part. The OPTIMAL team will select up to 30 eligible and interested patients for participation (or until data saturation is agreed by the researchers).[35 36] Saturation will be agreed when no additional themes emerge from the data.[36] Purposive sampling[37] will be used to ensure maximal diversity in the characteristics of the participants in terms of age, sex, ethnicity, socioeconomic status and number and types of long-term health conditions.

HCPs will be recruited through known contacts and British HCP bodies. This will include the British Geriatric Society, Royal College of General Practitioners, Society for Academic Primary Care, Royal College of Physicians, National Pharmacy Association, and the Royal College of Nursing. We will contact these organisations and request they send a letter or email describing the study to all eligible HCPs (online supplemental material). Any HCP interested in taking part will be instructed to email or phone the OPTIMAL research team directly. We aim to recruit until data saturation[36] is reached, up to a maximum of 30 HCPs using purposive sampling based on profession, age, years of experience, sex, ethnicity and place of practice (urban/rural, teaching/non-teaching, etc).

## Study materials

To encourage a rich dialogue during these interviews, a case vignette[38] (online supplemental material) will be used in conjunction with a semistructured topic guide (one for patients and one for HCPs; online supplemental material). The topic guides will explore patients' and HCPs' attitudes towards AI-informed clinical decision-making compared with clinician-directed decision-making, the perceived advantages and disadvantages of both methods and attitudes to risk management. These topic guides were developed based on the literature and guidance from the OPTIMAL PPAG.

The case vignettes were codeveloped by clinicians within the OPTIMAL team and the PPAG to reflect an expected decision-making process for a patient presenting with a common combination of multiple health conditions with both physical and mental comorbidities. HCPs will be asked how their decision-making process compares to potential outputs of the AI-based tool, whereas patients will be asked if the vignette's trajectories fit with their owned lived experiences in terms of the process for receiving their diagnoses and prescriptions. Participants will be encouraged to elaborate on their answers using prompts, probes and follow-up questions. The topic guide questions and case vignette were piloted with two patients and one HCP. Following these pilot interviews, the length of the case vignettes was shortened as it was said to be too long during the pilot; no other changes were made.

## Data collection

The SSIs will be conducted virtually via Zoom or Teams, by telephone or face-to-face[39 40] according to participant preference. There will be two interviewers; one will conduct the interview with patients and one will interview HCPs. All interviews will be audio recorded. Age, sex, ethnicity and number of existing conditions will be collected from the patients. Age, sex, ethnicity, profession and years of experience will be collected from the HCPs who take part. Prior to the interview, all participants will be provided with a summary explanation of AI (online supplemental material) in general and healthcare contexts.

## Data analysis

Interviews will be transcribed verbatim. NVivo (V.10 for Windows) will be used for data management. The deidentified transcripts generated from the recordings will be passed through word protectors and will be stored with access granted only for the data analysing team, allowing broad perspectives on the data. An inductive thematical analysis will initially be conducted on both HCP and patient transcripts whereby codes will be assigned line-by-line using the Framework Method.[41] Interviews will be analysed concurrently and iteratively to inform a reflexive process and create a cycle of data collection, analysis and

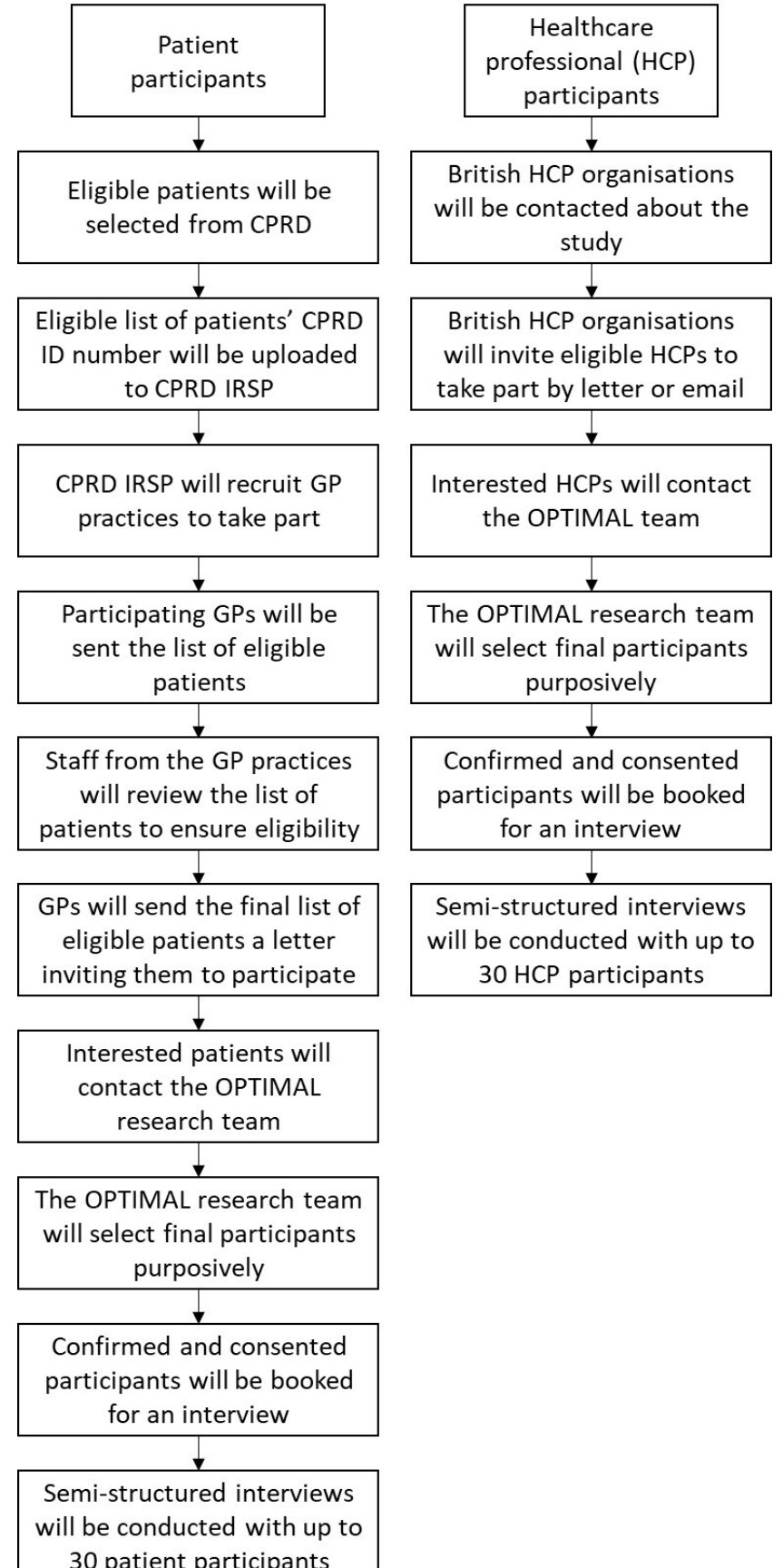

**Figure 2** Flowchart of study participant recruitment. CPRD, Clinical Practice Research Datalink; GP, general practitioners; IRSP, interventional research service platform; OPTIMAL, OPTIMising therapies, disease trajectories and Artificial intelligence (AI) assisted clinical management for patients Living with multiple long-term health conditions.

planning what questions to add or amend in subsequent interviews. Similar codes will be combined and themes will be identified in an analytical framework. Ten percent of HCPs and patient transcripts will be coded by a second researcher who did not facilitate the interviews; findings will be compared, and any disparities will be resolved through discussion with the wider team. After reviewing and revising, the final themes will be determined and the interpretations explored. A summary of overall themes will be sent to patient and HCP participants for comment.

A further analysis of HCP transcripts will be carried out using Buck *et al*'s theoretical framework as a starting point for a deductive analysis to explore and separate out participants' affective, cognitive and behavioural components of their attitudes towards AI use in managing multiple health conditions. Codes will be assigned to Buck *et al*'s framework to see if and in what ways the data fit the model and deepen understanding of attitudes towards AI. This framework has not previously been applied to patient perspectives; thus, we do not intend to apply it a priori to the patient transcripts.

As the data coding framework and overall themes emerge these will be discussed with the PPAG and members of the multidisciplinary research team to establish any further areas for exploration or clarification in subsequent interviews. A list of overall and individual themes for patients, and HCPs will be compiled to allow for cross group/individual and purposive sampling characteristics comparison and to understand how individual characteristics of HCPs (eg, age and prior experience with AI) and environmental influences (eg, current working environment, media influences, available technology in the workplace) may influence perspectives.

### Patient and public involvement

Members of the public and patients with experience of multiple health conditions have been involved in every stage of the OPTIMAL study. A PPAG was set up at the inception stage of the study comprising eight people who have lived experience of multiple health conditions (either directly or as a carer). We ensured that equality, diversity and inclusivity were prioritised and people of varied age, ethnicity, sex, geographic location in the UK and experience with multiple health conditions were invited to take part. Our PPAG contributed to the development of the study including objective setting, design and study materials through regular meetings. The topic guide, participant information document, consent form and invitation letter for patients were all reviewed by and improved according to feedback from our PPAG before ethical approval was sought. Throughout the study, meetings are expected to take place every 3 months with the PPAG. It is anticipated that these meetings will be particularly useful during the recruitment, data collection and analyses stages of the qualitative component this protocol describes to ensure the study aims and objectives are being met. To date, the impacts of the PPAG have included expanding the range of HCPs invited to take part (ie, to include pharmacists), improved the inclusiveness of the recruitment strategy and amending the design of the case vignettes to be more representative of their experiences.

### ETHICS AND DISSEMINATION

Ethical approval for the study was obtained from National Health Service (NHS) Research Ethics Committees (REC) (Reference: 22/SC/0210). Ethical approval to use the anonymised data from the CPRD dataset to select the eligible participants was obtained from the CPRD Expert Review Committee and the Central Advisory Committee (Reference:21_000683).

GPs will not know who decides to participate from the eligible patients they contact for recruitment and likewise, none of the HCP bodies will know which members decide to participate. Written informed consent will be obtained from the face-to face interviews and either electronically completed written consent forms or audio informed consent will be collected from the video or phone interviews; in the latter scenario, researchers will produce a written version of the consent on their behalf. If a participant withdraws from the study within 2 weeks of the interview, data collected from the participant will be securely destroyed. Otherwise, all data provided from consented participants will be used in the final analysis.

Data collection began with HCPs in October 2022 and are expected to be completed by September 2023; data collection with patients is expected to begin in August 2023 and completed by November 2023; analyses will be completed by February 2024. The final study report will be circulated to the relevant stakeholders and the summary of the final report will be available to the public by the National Institute for Health and Care Research (NIHR) and accompanied by a plain language summary. Furthermore, the study findings will be shared through peer-reviewed publications, public engagement activities and national or international conferences.

### DISCUSSION

Programming an effective and accurate AI-based decision-making tool using advanced machine learning and bioinformatics algorithms is one way to improve our understanding of how to reduce complications and additional comorbidities and safely manage polypharmacy in patients with multiple health conditions. However, patients and HCPs may feel there are barriers, risks and challenges for the use of AI in healthcare as well as benefits. Understanding these perceived barriers and potential risks is vital for effective implementation. This qualitative study will capture patients and HCPs perspectives on an AI-informed decision-making tool developed as part of the OPTIMAL study.

A large proportion of people aged 65 years or older have multiple health conditions.[42] AI-based tools have great potential in optimising disease management in the context of multiple health conditions and polypharmacy,

where the use of clinical trials are greatly limited. A major strength of this study is the use of data from the CPRD Aurum database to select eligible patients. CPRD Aurum is representative of the UK general population in terms of geographic spread, age, sex, deprivation and patterns of diseases.[43] This will facilitate the selection of a diverse sample of patients with multiple health conditions in terms of known population demographics. Another strength is the study's dedicated PPAG; they have and will continue to play an important role at every stage of the study ensuring that this research and the AI-based tool are of value to patients and carers with multiple health conditions. This also provides a unique opportunity for open dialogue between patients and members of the public, researchers and clinicians so they understand each other's perspectives and can discuss distinct challenges and learn from each other. By conducting qualitative interviews, this study will provide an in-depth understanding of patient and HCP perspectives which is vital for optimising the development and future implementation of an effective AI tool.[18] The systematic methodology of the Framework Method, input from the PPAG and respondent validation, will increase the trustworthiness of the conclusions while reducing researcher bias.[44] A limitation of the study to note is the non-probability sampling and involving patients only from the West Midlands which may limit the generalisability of the findings to patients from other regions in England, however the high socio-demographic diversity[45] of the area will aid understandings about how perceptions may vary between different demographic groups.

To conclude, our study will elucidate the perceptions of HCPs caring for patients with multiple health conditions and patients living with multiple health conditions on the use of AI-based clinical decision support for the management of multiple health conditions. AI-based tools have the potential to improve the care of multiple health conditions through better prescription management and deprescription; thus, reduce polypharmacy, side effects, drug burden and ultimately excess morbidity and premature mortality. It is therefore vital for patients and HCPs to raise their concerns and potential barriers and facilitators regarding the use of an AI-based tool for the tool to effectively work during the shared decision-making process. Our findings will enable a better understanding on how an AI-based tool can be effectively developed and implemented in a way that is acceptable, trustworthy and allows effective use by HCPs and is acceptable and accessible to the patient groups who are likely to benefit the most from the tool .

**Acknowledgements** The authors would like to thank the members of the PPAG (currently comprised of: Robert Jasper, Christine Michael, Jenny Negus, Gillian Richards, Lynne Wright and Janice Connelly) for their invaluable contribution to the conceptualisation and development of this study thus far.

**Contributors** In collaboration with the PPAG, all authors contributed to the conceptualisation of the study. JC and SF drafted the study tools, led the ethics application and piloted the topic guides and case vignettes with supervision from FC, TJ, KN and SG. NJG and TEG drafted all versions of the manuscript with supervision from FC. TJ, KN and SG are senior authors overseeing the project.

**Funding** This work is independent research funded by the National Institute for Health and Care Research (NIHR) (OPTIMising therapies, disease trajectories, and AI assisted clinical management for patients Living with complex multimorbidity (OPTIMAL study), NIHR202632). The views expressed in this publication are those of the author(s) and not necessarily those of NIHR or The Department of Health and Social Care.

**Competing interests** None declared.

**Patient and public involvement** Patients and/or the public were involved in the design, or conduct, or reporting or dissemination plans of this research. Refer to the Methods section for further details.

**Patient consent for publication** Not applicable.

**Provenance and peer review** Not commissioned; externally peer reviewed.

**ORCID iDs**
Niluka Jeewanthi Gunathilaka http://orcid.org/0000-0001-6887-8790
Tiffany E Gooden http://orcid.org/0000-0002-3905-5477
Shamil Haroon http://orcid.org/0000-0002-0096-1413
Francesca Crowe http://orcid.org/0000-0003-4026-1726
Krishnarajah Nirantharakumar http://orcid.org/0000-0002-6816-1279
Sheila Greenfield http://orcid.org/0000-0002-8796-4114

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
