## [Reviewer comments · BMJ Open]

ARTICLE DETAILS

TITLE (PROVISIONAL)	Perceptions on artificial intelligence-based decision making for coexisting multiple long-term health conditions: protocol for a qualitative study with patients and healthcare professionals
AUTHORS	Gunathilaka, Niluka; Gooden, Tiffany; Cooper, Jennifer; Flanagan, Sarah; Marshall, Tom; Haroon, Shamil; D'Elia, Alexander; Crowe, Francesca; Jackson, T; Nirantharakumar, Krishnarajah; Greenfield, Sheila

VERSION 1 – REVIEW

REVIEWER	Sinnott, Carol University of Cambridge Department of Public Health and Primary Care
REVIEW RETURNED	30-Aug-2023

GENERAL COMMENTS	Page 6, line 29 – what does developed mean in this context? Page 6, line 40- “predicting the next health condition that people with multiple health conditions might develop” – can you clarify how this information might be used e.g. screening/diagnostic investigations/mitigation strategies? It would be useful to have more description of the eligibility criteria for patients – is it simply two or more conditions? Is a validated list of conditions being used to search CPRD? Are risk factors like dyslipidaemia or hypertension being included as conditions? If the threshold for inclusion is set low, such as two or more conditions and/or risk factors are included as “conditions”, the potentially eligible population could represent the majority of adult patients in the GP attending population- what impact will this have on your data? Is there merit to using a higher threshold for multimorbidity (eg three +conditions, or 10+ medications, or complexity otherwise defined?) From all eligible patients attending a participating practice, how will the 30 to be invited be chosen and by whom? Patients who chose not to share their GP data will have no chance to participate in this research – but their views on AI are important. Given the proportion of patients signing data opt outs is rising, what impact will this have on your findings? Page 9, line 9- have you considered asking HCPs working in the participating practices to do interviews? Figure 2 is not labelled.
--

REVIEWER	Martinho, Andreia Tufts University
REVIEW RETURNED	22-Sep-2023

GENERAL COMMENTS	Patients' and healthcare professionals' perceptions on artificial
---

	intelligence based decision making for coexisting multiple long-term health conditions: A qualitative study protocol This study protocol manuscript relates to ongoing research that aims to investigate the perceptions of patients and healthcare professionals on AI-based decision-making for coexisting multiple long-term health conditions. This is an important and timely topic. The authors should consider the following suggestions to improve the study protocol: 1. Title: The use of apostrophe after (two) plural nouns is a bit controversial in a scientific publication. Perhaps consider changing the title to "Perceptions of patients and healthcare professionals on ...". 2. Introduction In general, the Introduction needs some improvements:  - The definition of "AI decision-making tools" is unclear - please provide more background and examples. - "There were mixed opinions about the impact that AI may have on the clinician-patient interaction" - which opinions are these? Please inform the reader by providing a summary of the findings reported in the literature. - " it has been difficult to harness informed insights from patients and clinicians about the use of AI in decision-making." - In general, the literature review section needs to be substantially improved, as the authors have failed to include several important articles that have addressed similar research questions (Patients: e.g. "A survey of pregnant patients' perspectives on the implementation of artificial intelligence in clinical care"; Medical Practitioners: "A healthy debate: Exploring the views of medical doctors on the ethics of artificial intelligence"). 3. Theoretical Framework It seems that the authors will use the work of Buck et al so it is not clear why they describe Rogers' Diffusion Theory. 4. Methods and Analysis The study could be improved by including also a question(s) about liability and medical standard of care in the AI paradigm.
--	---

REVIEWER	Aquino, Yves Saint James University of Wollongong, Australian Centre for Health Engagement, Evidence and Values
REVIEW RETURNED	27-Sep-2023

GENERAL COMMENTS	Thanks for the opportunity to review the protocol "Patients' and healthcare professionals' perceptions on artificial intelligence-based decision making for coexisting multiple long-term health conditions" (ID bmjopen-2023-077156). The protocol is a great initiative in ensuring engagement with health consumers and health service workers. Below are my comments and suggestions to improve the protocol for publication.
---

	Major concerns:  1. Conflict of interest: Please make clear that the same researchers conducting this study are the same as those developing the OPTIMAL AI tool. This is a clear conflict of interest, and I imagine that it's highly possible to oversell the AI tool to increase likelihood of support. Please provide strategies to ensure that researchers will be impartial in conducting the interviews. The introduction and abstract should make clear that this study is part of a larger study that aims to develop an AI tool. 2. Introduction section: The protocol needs to provide review of existing AI products intended for healthcare broadly, and/or multiple chronic conditions specifically. 3. Theoretical framework: Please justify why the theoretical frameworks will only be used in the analysis stage? Why not use it in developing the questionnaire or in recruiting participants? Minor concerns:  1. In the Intro section (p. 5, line 54-): "AI is a term used to describe..." This phrase is misleading as it initially looks like it's providing a definition of AI, but it ends up describing a specific application. Please provide AI definition from literature. 2. Please justify exclusion of patients with a terminal diagnosis (page 8, line 18-20). 3. Please clarify the CPRD's policy on whether patients consent to be contacted for possible participation to a research study (page 8). 4. Please justify how you arrived at 30 participants for patient and HCP cohorts. Please clarify strategies or criteria to determine you have reached data saturation. 5. Please clarify whether analysis will only be initiated upon completion of all interviews and encoding of transcripts. 6. I'm confused by your use of two frameworks (Framework Method plus Buck et al's Framework). You can just say you're doing both inductive and deductive analysis, and you're guided initially by Buck et al.'s. 7. Please state the number of coders, if coding will be based on consensus, and strategies to address conflicts.
--	--

VERSION 1 – AUTHOR RESPONSE

Reviewer 1 comments:

Comment 1

Page 6, line 29 – what does developed mean in this context?

Response 1

Apologies for the confusion, we have removed the word 'developed' from this sentence.

Comment 2

Page 6, line 40- "predicting the next health condition that people with multiple health conditions might develop" – can you clarify how this information might be used e.g. screening/diagnostic investigations/mitigation strategies?

Response 2

We envision that an AI tool capable of making predictions on what health conditions people are at risk for could be used to inform future screening programmes, investigations, mitigation strategies and treatment plans. We have amended the sentence to include this information, as below (lines 39-43).

“The OPTIMising therapies, disease trajectories, and artificial intelligence (AI) assisted clinical management for patients Living with multiple long-term health conditions (OPTIMAL) study aims to produce an AI-based tool to be used in primary care settings for planning the best treatment strategies and predicting the next health condition that people with multiple health conditions might develop to inform screening, investigations, prevention and/or treatment.”

Comment 3

It would be useful to have more description of the eligibility criteria for patients – is it simply two or more conditions? Is a validated list of conditions being used to search CPRD? Are risk factors like dyslipidaemia or hypertension being included as conditions? If the threshold for inclusion is set low, such as two or more conditions and/or risk factors are included as “conditions”, the potentially eligible population could represent the majority of adult patients in the GP attending population- what impact will this have on your data? Is there merit to using a higher threshold for multimorbidity (eg three +conditions, or 10+ medications, or complexity otherwise defined?)

Response 3

Patients were eligible for inclusion if they had two or more long-term conditions. This is consistent with the definition of multimorbidity established by the Academy of Medical Sciences (<https://acmedsci.ac.uk/file-download/39787360>). The list of conditions is based on a Delphi study by Ho et al. which identified 59 key conditions that are important to patients and research stakeholders for inclusion in research on patients with multiple long-term conditions (<https://www.ncbi.nlm.nih.gov/pmc/articles/PMC9978673/>). For interest, hypertension is included but hyperlipidaemia is not. We used the conditions recommended from this Delphi study as a baseline and discussed these within the OPTIMAL study team and our patient advisory group to determine a final group of 63 conditions for inclusion. We have now uploaded the list of conditions as supplementary material to the manuscript and referenced it in the manuscript (see below).

We agree with the reviewer, that inclusion of conditions such as hypertension, which are highly prevalent will mean that a very high proportion of adults are eligible. However, multi-morbidity is now the norm in adults over 60 and this does not negate the impact that even two co-existing conditions have on both life expectancy and quality of life and treatment burden (polypharmacy is especially relevant concerning hypertension) ([https://www.thelancet.com/journals/eclinm/article/PIIS2589-5370\(23\)00037-8/fulltext](https://www.thelancet.com/journals/eclinm/article/PIIS2589-5370(23)00037-8/fulltext)). We agree with the reviewer that insights from patients with higher complexity and disease/medication burden are likely to be particularly valuable. As mentioned in the manuscript (see below), we will use purposive sampling aiming to interview participants with a range of conditions and severity of multimorbidity (number of conditions and medications).

“This qualitative study will be conducted using SSIs (29) with two key groups of stakeholders. The first group will include people aged 18 years or older, with more than one long-term health condition who are registered with a GP in the West Midlands County in England that contributes to the Clinical Practice Research Datalink (CPRD) Aurum database. Sixty-three conditions were considered when assessing eligibility (Supplementary material); these were derived through existing evidence (30) and discussions with the OPTIMAL Patient and Public Advisory Group (PPAG).” (lines 109-114)

“Purposive sampling (37) will be used to ensure maximal diversity in the characteristics of the participants in terms of age, gender, ethnicity, socio-economic status and number and types of long-term health conditions.” (lines 146-148)

Comment 4

From all eligible patients attending a participating practice, how will the 30 to be invited be chosen and by whom?

Response 4

As mentioned in the manuscript in the 'Recruitment' section, eligible patients will be invited to participate by their GPs. The GPs will send an invitation letter and information sheet to eligible patients. Any patients interested in participating will be instructed to contact the OPTIMAL team by phone or email to register their interest. From the list of interested eligible patients, the OPTIMAL team will then purposively sample and contact up to 30 patients (until saturation is met) to arrange an interview. We have made slight amendments to the text to make this clearer (see below).

"Each practice who takes part will be provided a list of eligible patients together with their CPRD ID number through the IRSP dashboard within their practice. Once this list has been received by the practices, either the practice manager, nurse or GP will view the eligible patients' details. They will review their medical history to ensure they meet the eligibility criteria. Once eligibility is confirmed, these patients will be sent an invitation letter and participant information sheet (Supplementary material) from their GP. Interested patients will be asked to call or email the OPTIMAL research team, at which point they can ask any questions about the study.

To augment this recruitment method for patients and ensure we get a diverse sample, we will also circulate recruitment flyers through voluntary, third sector and patient support and community groups, including but not limited to Citizens UK (33) and Birmingham Voluntary Services Council,(34) and through help with from the OPTIMAL PPAG. Flyers will include information on eligibility and how to contact the OPTIMAL team to learn more about the study, ask questions, and receive a copy of the participant information sheet before deciding if they would like to take part. The OPTIMAL team will select up to 30 eligible and interested patients for participation (or until data saturation is agreed by the researchers).(35, 36) Saturation will be agreed when no additional themes emerge from the data.(36) Purposive sampling (37) will be used to ensure maximal diversity in the characteristics of the participants in terms of age, gender, ethnicity, socio-economic status and number and types of long-term health conditions." (lines 131-148)

Comment 5

Patients who chose not to share their GP data will have no chance to participate in this research – but their views on AI are important. Given the proportion of patients signing data opt outs is rising, what impact will this have on your findings?

Response 5

Thank you, this is a very important point. The proportion of patients signing data opt-outs is around 5% (<https://digital.nhs.uk/dashboards/national-data-opt-out-open-data>), and these patients may indeed differ from other patients regarding demographics, comorbidities and/or their perspectives of using an AI-based tool for managing clinical decisions. Since submission of the manuscript (June 2023), we have applied and received further ethics approval to augment our recruitment strategy for patients to include recruitment via social media and flyers; this would presumably overcome any responder bias linked to GP-based recruitment; this additional recruitment strategy has been added to the manuscript (see below).

"To augment this recruitment method for patients and ensure we get a diverse sample, we will also circulate recruitment flyers through voluntary, third sector and patient support and community groups, including but not limited to Citizens UK (33) and Birmingham Voluntary Services Council,(34) and through help with from the OPTIMAL PPAG. Flyers will include information on eligibility and how to contact the OPTIMAL team to learn more about the study, ask questions, and receive a copy of the participant information sheet before deciding if they would like to take part." (lines 138-143)

Comment 6

Page 9, line 9- have you considered asking HCPs working in the participating practices to do interviews?

Response 6

This was not considered due to financial, methodological and logistical reasons. For instance, we plan to recruit patients from 30 GP practices; we would have had to train at least 30 HCPs (one from each practice) on how to conduct qualitative research which would not be a practical approach due to project finances and timelines. The OPTIMAL study has a group of academic researchers, whom some are also qualified GPs, that were already trained and experienced in conducting qualitative research, familiar with the project and have protected time to conduct the interviews and analyse the data. It was also important to make patients fully understand the voluntary nature and anonymity of the study by ensuring them that their GP would not know whether they decide to participate or not and that their care will not change based on their decision to participate. This would have been compromised if HCPs working in the practices conducted the interviews.

Comment 7

Figure 2 is not labelled.

Response 7

The label of figure 2 (and figure 1) is included in the main text of the manuscript, after the references. If accepted for publication, we presume the production team will align the two. We are unsure why this didn't occur during the PDF production of our submission.

Reviewer 2 comments:

Comment 1

This study protocol manuscript relates to ongoing research that aims to investigate the perceptions of patients and healthcare professionals on AI-based decision-making for coexisting multiple long-term health conditions. This is an important and timely topic. The authors should consider the following suggestions to improve the study protocol:

Response 1

Thank you for your positive assessment of the protocol manuscript. We hope you find the amendments described below in response to your comments have elevated the paper for publication.

Comment 2

Title: The use of apostrophe after (two) plural nouns is a bit controversial in a scientific publication. Perhaps consider changing the title to "Perceptions of patients and healthcare professionals on ...".

Response 2

Thank you for this suggestion, we have amended the title to be as follows: Perceptions on artificial intelligence-based decision making for coexisting multiple long-term health conditions: A protocol for a qualitative study with patients and healthcare professionals.

Comment 3

Introduction. In general, the Introduction needs some improvements:

- The definition of "AI decision-making tools" is unclear - please provide more background and examples.

Response 3

It is important to note that within healthcare settings, AI has had limited practical application (<https://www.nejm.org/doi/full/10.1056/NEJMra2302038>). Existing technology has predominantly been

used for image-processing such as interpretation of electrocardiograms, retinal scans and radiology images. However, in primary care, the use of AI has, to date, been limited. We have amended the text in the manuscript to provide more background information on this (see below). Please also note that we included in our supplementary materials a description of AI that will be given to all participants to explain what AI is in general and in healthcare settings.

“Artificial intelligence (AI) tools are one solution that is being suggested to overcome some of the limitations in guidelines and evidence base for patients with multimorbidity.(10) AI is broadly defined as the science of “machines [that] do things that would require intelligence if done by people”.(11) Machine learning is a type of AI in which a computer with self-learning capacity can generate predictive algorithms and identify patterns from data.(12) This has been successful at tackling complex problems outside of healthcare;(10) however, the capacity to process large amounts of information, systematically and reliably, faster than the human brain means there is considerable potential for AI in complex healthcare decision making. The Royal College of General Practitioners has recognised that there is vast potential for AI in general practice.(13) There are some working examples to date in other settings (e.g.: secondary care, intensive care) where AI-based tools have been integrated for risk stratification and patient outcome optimisation,(14) but application of AI-tools in primary care is still in its infancy.(15, 16)” (lines 27-38)

Comment 4

- "There were mixed opinions about the impact that AI may have on the clinician-patient interaction" - which opinions are these? Please inform the reader by providing a summary of the findings reported in the literature.

Response 4

Apologies for the confusion. Please note as a response to the following comment (comment 5), we have reworded the text in the introduction regarding the existing literature (which included the sentence you raise in this comment). Please see response 5 below for the amended text.

Comment 5

- " it has been difficult to harness informed insights from patients and clinicians about the use of AI in decision-making." - In general, the literature review section needs to be substantially improved, as the authors have failed to include several important articles that have addressed similar research questions (Patients: e.g. "A survey of pregnant patients' perspectives on the implementation of artificial intelligence in clinical care"; Medical Practitioners: "A healthy debate: Exploring the views of medical doctors on the ethics of artificial intelligence").

Response 5

Thank you for this suggestion, we have updated the introduction of the manuscript to include further information from existing literature (see below). To keep the introduction concise and clear, we hope you agree that it is not practical to describe the details of all existing evidence, but instead provide an overview of existing literature. We hope you find our amendments have done just that.

“A 2021 systematic review of 23 qualitative studies exploring patient and public perspectives towards AI in the clinical setting found that the public were broadly positive about the concept of AI, but were concerned about the effectiveness of AI tools and felt that implementation should have human oversight.(17) However, in three-quarters of the included studies, discussions were based on AI as a hypothetical concept, rather than a real-world example (likely because few existed), which may limit the depth or specificity of the discussions.(17) Some qualitative studies have reported concerns from patients regarding the lack of a ‘human touch’ in using AI-technology, but most have optimism that AI-based tools could free up HCP time, leaving more time for patient interactions.(18) Studies of clinicians’ attitudes to AI are also generally positive, especially around the potential to take over

mundane administrative tasks and synthesise data.(18-21) However, studies exploring UK GPs' perspectives, including a large survey of 720 respondents, found that GPs believe human empathy, communication and tailoring to individual patients' values could not be replicated by AI.(21-23) Another study examining GP perspectives on using an AI-based documentation tool found they had concerns about the impact on their professional autonomy and who would bear the medico-legal responsibility for any outcomes.(19) Those in managerial and regulatory roles (including clinicians) have also expressed concerns over regulatory oversight of AI, the transferability/generalisability of the tools and the challenge of adopting AI within existing systems.(9) This was felt to be especially difficult within primary care which involves multiple independent practices and deals with a high burden of what were thought of as 'non-digitisable' healthcare problems such as mental health and chronic illness.(9)

Most existing studies exploring perspectives on AI have treated it as a broad and hypothetical concept.(17,18) Furthermore, limited evidence exist on stakeholder perceptions of AI-technology for managing multiple coexisting health conditions.(21) Exploration of current and informed perspectives using a real-world example of an AI-based tool is needed. Insights from HCPs regarding the challenge of implementing this technology in management of multimorbidity in primary care will be especially valuable." (lines 52-77)

Comment 6

Theoretical Framework. It seems that the authors will use the work of Buck et al so it is not clear why they describe Rogers' Diffusion Theory.

Response 6

Thank you for this feedback. We can see how the inclusion of text regarding Rogers' Diffusion Theory can cause confusion; as a response, we have removed this text from the amended manuscript.

Comment 7

Methods and Analysis. The study could be improved by including also a question(s) about liability and medical standard of care in the AI paradigm.

Response 7

These questions will not be considered as part of the current study protocol; however, as part of the OPTIMAL study group, further qualitative interviews are expected at a later stage to determine the optimal way to present information from the AI-based tool and determinants of preferences for future use. We will consider including questions related to liability and medical standard of care in these future interviews.

Reviewer 3 comments:

Comment 1

Dear editor and authors,

Thanks for the opportunity to review the protocol "Patients' and healthcare professionals' perceptions on artificial intelligence-based decision making for coexisting multiple long-term health conditions" (ID bmjopen-2023-077156). The protocol is a great initiative in ensuring engagement with health consumers and health service workers. Below are my comments and suggestions to improve the protocol for publication.

Response 1

Thank you for taking the time to review our protocol manuscript and providing helpful suggestions and feedback. We hope the amendments made described below have improved it the protocol.

Major concerns:

Comment 2

Conflict of interest: Please make clear that the same researchers conducting this study are the same as those developing the OPTIMAL AI tool. This is a clear conflict of interest, and I imagine that it's highly possible to oversell the AI tool to increase likelihood of support. Please provide strategies to ensure that researchers will be impartial in conducting the interviews. The introduction and abstract should make clear that this study is part of a larger study that aims to develop an AI tool.

Response 2

The reviewer has raised an important point about ensuring there are strategies for maintaining impartiality when conducting the interviews. The first set of interviews with people with multiple long-term health conditions and health professionals (of which this protocol describes), will help develop an understanding of the perceptions in the use of AI tools in healthcare. There will be a second set of interviews at a later stage in the OPTIMAL study that will be used to develop the options for the characteristics and features of the AI tool that will be used for a discrete choice experiment. We do not intend to garner support for the AI tool. Furthermore, the researchers involved in recruitment, data collection and data analysis for the qualitative study are not the same researchers who will be developing the AI-based tool. To address the reviewer's comment, we have added text regarding this within the competing interest section (see below).

"The researchers involved in recruitment, data collection and data analysis for the qualitative study are not the same researchers who will be developing the AI-based tool." (lines 305-307)

Comment 3

Introduction section: The protocol needs to provide review of existing AI products intended for healthcare broadly, and/or multiple chronic conditions specifically.

Response 3

Please see response 3 to comment 3 by reviewer 2.

Comment 4

Theoretical framework: Please justify why the theoretical frameworks will only be used in the analysis stage? Why not use it in developing the questionnaire or in recruiting participants?

Response 4

Thank you for this helpful comment. This is a relatively new area of research, given that the application of AI in primary care settings is largely hypothetical at present. Therefore, we did not feel it was appropriate to overly constrain the work by introducing theory, and therefore a particular slant from the outset. We have clarified this in the "Theoretical framework" section of the manuscript (see below). We want to see what freely emerges inductively from the data, but to use Buck's theory to explore and separate out the participants' affective, cognitive and behavioural components of their attitudes toward AI, if the emerging data does fit this model.

"Given that the application of AI in general practice is largely hypothetical at present, current attitudes to AI from potential users are critical to the development of technologies that could then be implemented in practice. Therefore, we did not feel it appropriate to overly constrain the work by introducing theory, and therefore a particular slant, from the very outset. However, we will apply Buck et al.'s theoretical lens at the analysis stage. This lens will be used as a framework to further explore the data from HCP interviews to provide a broader conceptual understanding of how the affective, cognitive and behavioural components of attitude determine how HCPs evaluate their views on AI in managing multiple health conditions." (lines 93-101)

Minor concerns:

Comment 5

In the Intro section (p. 5, line 54-): "AI is a term used to describe..." This phrase is misleading as it initially looks like it's providing a definition of AI, but it ends up describing a specific application. Please provide AI definition from literature.

Response 5

Please see response 3 to comment 3 by reviewer 2.

Comment 6

Please justify exclusion of patients with a terminal diagnosis (page 8, line 18-20)

Response 6

The main reason for excluding patients with a terminal diagnosis (those expected to be in the last 12 months of life) was to avoid causing any undue distress these patients by asking about their health and healthcare. We have added this to the manuscript (see below).

"Patients with a terminal diagnosis (prognosis of less than 12 months) will be excluded to avoid undue distress as well as those deemed by their GP not to have the capacity to consent or participate." (lines 115-117)

Comment 7

Please clarify the CPRD's policy on whether patients consent to be contacted for possible participation to a research study (page 8).

Response 7

Only records from patients that have not opted out of allowing their data to be used for research and planning purposes are uploaded to CPRD from participating GPs. CPRD has ethics approval from the Health Research Authority for collecting anonymised medical records and completes an annual NHS Data Security and Protection Toolkit assessment to demonstrate that it meets the required standard for holding data securely. More information on the process can be found on their website (<https://cprd.com/safeguarding-patient-data>). Thus, only patients that do not opt out of sharing of their data will be identified via CPRD. Written informed consent will be obtained before any interviews commence. We have added some text in the manuscript to ensure this is clear.

Comment 8

Please justify how you arrived at 30 participants for patient and HCP cohorts. Please clarify strategies or criteria to determine you have reached data saturation.

Response 8

For pragmatic purposes, we have funded for up to 30 patient participants and 30 HCP participants. However, this is based on assumptions (given existing literature and guidance on qualitative methods) that data saturation will be met with this maximum sample size. Data analysis will occur in parallel to data collection; thus, data saturation will be determined by the study team when no additional codes or themes are formed from the data. Data collection will be complete once data saturation has been achieved from patient participants and separately for HCP participants. Please see below the text regarding this in the manuscript.

"The OPTIMAL team will select up to 30 eligible and interested patients for participation (or until data saturation is agreed by the researchers).(35, 36) Saturation will be agreed when no additional themes emerge from the data.(36)" (lines 143-146)

"We aim to recruit until data saturation (36) is reached, up to a maximum of 30 HCPs using purposive sampling based on profession, age, years of experience, sex, ethnicity and place of practice (urban/rural, teaching/non-teaching etc)." (lines 154-157)

Comment 9

Please clarify whether analysis will only be initiated upon completion of all interviews and encoding of transcripts.

Response 9

The interviews and encoding of transcripts will take place concurrently and iteratively; this will allow for any new topics or themes to be explored in subsequent interviews. We have added this to the manuscript (see below).

“Interviews will be analysed concurrently and iteratively to inform a reflexive process and create a cycle of data collection, analysis and planning what questions to add or amend in subsequent interviews.” (lines 188-190)

Comment 10

I'm confused by your use of two frameworks (Framework Method plus Buck et al's Framework). You can just say you're doing both inductive and deductive analysis, and you're guided initially by Buck et al.'s.

Response 10

Thank you for highlighting an area in need of clarification. We will use the Framework Method to guide the initial analysis process using the inductive technique. We will then apply Buck et al.'s, theoretical framework just to HCP transcripts as a subsequent deductive analysis; therefore, these are two separate analyses. We have amended the manuscript to make this clearer (see below).

“Interviews will be transcribed verbatim. NVivo (version 10 for Windows) will be used for data management. The de-identified transcripts generated from the recordings will be passed through word protectors and will be stored with access granted only for the data analysing team, allowing broad perspectives on the data. An inductive thematic analysis will initially be conducted on both HCP and patient transcripts whereby codes will be assigned line-by-line using the Framework Method.(41) Interviews will be analysed concurrently and iteratively to inform a reflexive process and create a cycle of data collection, analysis and planning what questions to add or amend in subsequent interviews. Similar codes will be combined and themes will be identified in an analytical framework. Ten percent of HCP and patient transcripts will be coded by a second researcher who did not facilitate the interviews; findings will be compared, and any disparities will be resolved through discussion with the wider team. After reviewing and revising, the final themes will be determined and the interpretations explored. A summary of overall themes will be sent to patient and HCP participants for comment.

A further analysis of HCP transcripts will be carried out using Buck et al.'s theoretical framework as a starting point for a deductive analysis to explore and separate out participants' affective, cognitive and behavioural components of their attitudes toward AI use in managing multiple health conditions. Codes will be assigned to Buck et al.'s framework to see if and in what ways the data fits the model and deepens understanding of attitudes toward AI. This framework has not previously been applied to patient perspectives; thus, we do not intend to apply it a priori to the patient transcripts.” (lines 183-202)

Comment 11

Please state the number of coders, if coding will be based on consensus, and strategies to address conflicts.

Response 11

For HCPs all interviews and coding will be conducted by the primary researcher (JC). To improve validity of the coding framework a second researcher (SF) will independently code 10% of the

transcripts and the findings will be compared. Any disparities will be discussed with the entire study research team. Similarly, patient interviews will be conducted and coded by the primary researcher (SF) and 10% of the transcripts will be coded by a second researcher (JC) to compare findings. This has been added to the manuscript (see below).

“Ten percent of HCP and patient transcripts will be coded by a second researcher who did not facilitate the interviews; findings will be compared, and any disparities will be resolved through discussion with the wider team.” (lines 191-193)

VERSION 2 – REVIEW

REVIEWER	Sinnott, Carol University of Cambridge Department of Public Health and Primary Care
REVIEW RETURNED	02-Nov-2023
GENERAL COMMENTS	Thanks to the authors for their clear, well-laid out responses. Of note, my question "Page 9, line 9- have you considered asking HCPs working in the participating practices to do interviews?" was misunderstood by the team. I did not mean had they considered asking the HCPs to conduct the interviews but to be interviewed!
REVIEWER	Martinho, Andreia Tufts University
REVIEW RETURNED	21-Nov-2023
GENERAL COMMENTS	The authors have addressed the pressing issues that were raised by this reviewer. The study protocol meets the standards for publication. As a final note, this reviewer suggests the authors consider removing the definition of AI, which is quite dated ("machines [that] do things that would require intelligence if done by people"(11)) in light of the current machine learning data-driven AI.
REVIEWER	Aquino, Yves Saint James University of Wollongong, Australian Centre for Health Engagement, Evidence and Values
REVIEW RETURNED	17-Nov-2023
GENERAL COMMENTS	I am happy to recommend publication. My concerns were adequately addressed.